# Optimal Design of Voltage Equalization Ring for the 1100 kV DC Voltage Proportional Standard Device Based on the Nation Standard Device Neural Network and Grey Wolf Optimization Algorithm

**Wanjun Zhu** [1,2]**, Yin Gao** [3]**, Liang Qin** [1,2,*]**, Yuqing Duan** [3]**, Zhigang Bian** [3]**, Min He** [1,2] **and Kaipei Liu** [1,2]

1   Hubei Key Laboratory of Power Equipment & System Security for Integrated Energy, Wuhan 430072, China; 2018302070163@whu.edu.cn (W.Z.); whuhemin@whu.edu.cn (M.H.); kpliu@whu.edu.cn (K.L.)
2   School of Electrical Engineering and Automation, Wuhan University, Wuhan 430072, China
3   State Grid Anhui Electric Power Company Limited, Hefei 230022, China; gaoy2015@ah.sgcc.com.cn (Y.G.); duanyq2979@ah.sgcc.com.cn (Y.D.); bianzg201x@ah.sgcc.com.cn (Z.B.)
*   Correspondence: qinliang@whu.edu.cn; Tel.: +86-189-8617-2977

**Abstract:** The DC voltage ratio standard device is an important tool for calibrating DC voltage transformers. At the 1100 kV voltage level, an increase in electric field intensity will increase the local heat generated inside the device, affecting the accuracy of its measurement. Using a suitable grading ring can even out the electric field intensity and reduce the maximum field strength to improve its measurement accuracy. This article mainly optimizes the design of the grading-ring structure of the 1100 kV DC voltage ratio standard device. First, a finite-element model of the 1100 kV DC voltage ratio standard device was built based on ANSYS; the electric field distribution around the voltage divider was calculated and analyzed, and a data set was constructed based on the calculation results. Secondly, for the optimization of electric field strength, this article presents the design of the nation standard device neural network, which learns the relationship between the structural parameters of the toroidal core and the field strength under the PyTorch 1.8 deep learning framework. Due to the strong convergence performance, few parameters, and ease of implementation of the grey wolf optimization algorithm, this study selected this algorithm to optimize the structural parameters of the grading ring. Finally, simulation examples are established in Python for validation. The experimental results indicate that the maximum field strength of the grading ring decreased from 12,161.1348 V/cm to 10,009.2881 V/cm, a reduction of 17.69%. The optimized structural parameters of the grading ring effectively reduced the electric field intensity around the 1100 kV DC voltage proportional standard device, providing a reliable and practical design approach for the selection of the DC voltage ratio standard device.

**Keywords:** DC voltage ratio standard device; finite-element method (FEM); NSD neural network; grey wolf optimization algorithm; optimization of grading ring



## 1. Introduction

The uneven distribution of energy resources and loads in China determines that the pattern of "transmitting electricity from the west to the east" and "transmitting electricity from the north to the south" will not change in the next 40 years [1]. The development of high-voltage direct current (HVDC) transmission can achieve flexible and controllable, low-loss, and highly reliable long-distance transmission across regions throughout China. It will provide assurance for large-scale optimization and mutual compensation of resources, making the future role of HVDC transmission even more crucial in the strategic development of China's energy [2]. Accurate measurement of high DC voltage is the foundation for ensuring the safe, stable, and economical operation of DC transmission systems, holding

significant importance for safeguarding China's energy security [3]. Establishing a national DC voltage proportional standard device to achieve self-calibration of values and conducting on-site calibration tests to achieve accurate transmission of values are effective means to ensure that DC voltage proportional values are accurate and reliable [4]. Currently, China has put into operation the world's highest-voltage-level, highest-transmission-capacity, and longest-transmission-distance DC transmission project—the Changji-Guquan $\pm$ 1100 kV DC transmission project. However, a high-precision 1100 kV DC voltage ratio standard device has not been established, and the corresponding on-site transmission technology for the 1100 kV DC voltage ratio standard device is not yet mature. This has led to the inability to trace the 1100 kV DC voltage ratio standard device, impacting the safety of the system operation [5,6].

The DC voltage ratio standard device adopts the resistive voltage division principle [7]. Therefore, during the development of the standard device, three main influencing factors are primarily considered: first, the variation in resistance value due to changes in environmental temperature and the self-heating of resistance components; second, corona discharge caused by high voltage; and third, the influence of leakage current along the insulation support [8]. In 2015, China established the national standard for 1000 kV DC voltage ratio. However, with the highest voltage level of China's UHVDC (ultra-high voltage direct current) transmission projects reaching 1100 kV, there is an urgent need to establish a 1100 kV DC voltage standard [9]. The electric field strength and distribution around the DC voltage ratio standard device affect the local heating of resistors at rated voltage. As the voltage level increases, the field strength distribution becomes more uneven, leading to greater local heating. This, in turn, causes significant changes in the resistance values of internal resistor components, impacting the accuracy of the voltage divider. In severe cases, it may result in thermal breakdown of resistor components [10]. The structural parameters of the grading ring directly influence the magnitude and distribution of the electric field around the DC resistor standard voltage divider. Installing an optimally designed grading ring can effectively improve the potential and electric field distribution, reduce the variation in resistance values of internal resistor components and insulation resistors within the voltage divider, and enhance the measurement accuracy of the DC standard resistor voltage divider.

In the past, many researchers have devoted themselves to the optimization design of grading ring structures, employing various methods including exhaustive methods, orthogonal methods, neural networks, and genetic algorithms [11–15]. Reference [11] continuously adjusted the grading ring structure parameters in the computational model, comparing the electric field intensity with and without the installation of the grading ring and varying the size and position of the grading ring. However, it did not use optimization methods. Reference [12] took the potential distribution of insulator strings as the optimization target for grading-ring size and position, calculating the maximum voltage borne by a single insulator under different size and position combinations to determine the optimal parameters for the grading ring. However, the exhaustive search method suffers from low efficiency, and in this literature the large step size used in the exhaustive search method makes it difficult to find the optimal parameters. Reference [13] proposed an insulation grading-ring parameter optimization method based on an improved genetic algorithm, with the maximum surface electric field along the insulator as the objective function. They established a mathematical model for the optimization of grading-ring parameters to improve the electric field intensity on the insulator surface, achieving a good improvement in the electric field intensity. Reference [14] applied the orthogonal experimental method of multi-objective optimization to optimize the size parameters and installation position of the grading ring. However, this method may lack precision in the optimization of the grading ring. Reference [15] used an artificial neural network to fit the relationship between grading-ring structural parameters and the maximum surface electric field, obtaining optimized structural parameters. Nevertheless, there is still room for further optimization of the obtained parameters. Most optimization designs

for the grading ring are not precise enough, often relying on exhaustive methods and orthogonal experimental methods. However, using neural networks can provide more accurate optimization parameters, resulting in a greater improvement in the distribution and maximum strength of the electric field.

Regarding the above-mentioned issue, the work of this paper is divided into the following three points:

1. The construction of a finite element model of the 1100 kV DC voltage ratio standard device using ANSYS simulation software 2022 R1, and the calculation and analysis of its surrounding electric field distribution.
2. The proposal of a neural network named NSD based on the results of the electric field calculation to fit the relationship between the structural parameters of the grading ring and the electric field intensity.
3. The utilization of the grey wolf optimization algorithm to optimize the structural parameters of the grading ring. The optimal selection of the grading ring's structural parameters significantly improves the size and distribution of the electric field around the DC voltage ratio standard device, thereby enhancing the measurement accuracy of the standard device. The aforementioned work provides guidance for the establishment of the 1100 kV DC voltage ratio standard device.

## 2. Establishment of the Simulation Computational Model for the 1100 kV DC Voltage Ratio Standard Device

Firstly, we utilized ANSYS simulation software to construct a finite element model of the 1100 kV DC voltage ratio standard device, and calculated and analyzed its surrounding electric field distribution. In ANSYS, a simulation model is constructed using the finite element method to provide data support for the standard device design [16]. Through the finite element method, we successfully transformed the infinite domain problem into a finite domain problem. Therefore, for the 1100 kV standard device, the solutions for the electric field and potential distribution can ultimately be understood by addressing the following issues.

In the solution domain:

$$\nabla^2 E = 0, \tag{1}$$

At different media and continuous interfaces:

$$n(D_1 - D_2) = 0, \tag{2}$$

At the potential on the conductor side:

$$V_1 = V_0, \tag{3}$$

At the potential along the finite boundaries:

$$V_2 = 0, \tag{4}$$

In the equation, $E$ represents the electric field strength within the region to be solved, $n$ is the normal direction perpendicular to the interface of the media, $D_1$ and $D_2$ are the electric potential displacements perpendicular to the interface on either side of the boundary; $V_o$ is the specific potential value on the conductor side, $V_1$ is the potential on the conductor side, and $V_2$ is the potential along the finite boundary. To ensure the accuracy of the simulation analysis and reduce computational time, the solution domain is divided into three regions. Theoretical analysis indicates that the electric field concentration is highest in the central region, requiring the highest precision. Therefore, special attention is given to the core region, which includes the standard device and the grading ring, and it is densely meshed. Conversely, the other two regions are sparsely meshed according to the principle of gradually decreasing density with distance from the center.

The 1100 kV DC voltage ratio standard device is internally divided into an inner-shielding resistor layer, a measuring resistor layer, and an outer-shielding resistor layer. Each resistor layer is composed of series-connected resistors distributed uniformly in a helical pattern along the insulation tube from the top to the bottom of the standard device. Therefore, the standard device is not a strictly three-dimensional axisymmetric structure. However, at each resistor node position the measuring resistor and the shielding resistor are in parallel, and the resistance values of the measuring resistor and the shielding resistor are the same for each layer; the potentials of the three resistor layers are very close. When the inner-shielding resistor layer forms equipotential shielding with the measuring resistor layer, it significantly reduces the leakage current along the insulating support of the measuring resistor layer. Due to its minimal impact on the electric field distribution, the inner-shielding resistor layer was ignored in the electric field simulation model to simplify calculations, with only the measuring resistor layer and outer-shielding resistor layer considered. As both the measuring resistor layer and outer-shielding resistor layer have a uniform distribution of resistance from top to bottom, to further simplify the model they are replaced by two cylinders with equal resistance distribution. In the end, the 1100 kV standard device is simplified into a three-dimensional axisymmetric structure. When constructing the ANSYS finite element model for a three-dimensional axisymmetric structure, modeling can be carried out by taking half of the main axis profile in two dimensions. This approach simplifies the three-dimensional electrostatic field problem into a two-dimensional electrostatic field problem. In this model, voltage is used as the degree of freedom for solving each element node. When conducting electrostatic field calculations, a DC high voltage is applied through a high-voltage conducting rod to the top of the measuring resistor layer and the outer-shielding resistor layer, with the bottom grounded. The constructed finite-element model of the 1100 kV DC voltage ratio standard device is shown in Figure 1.

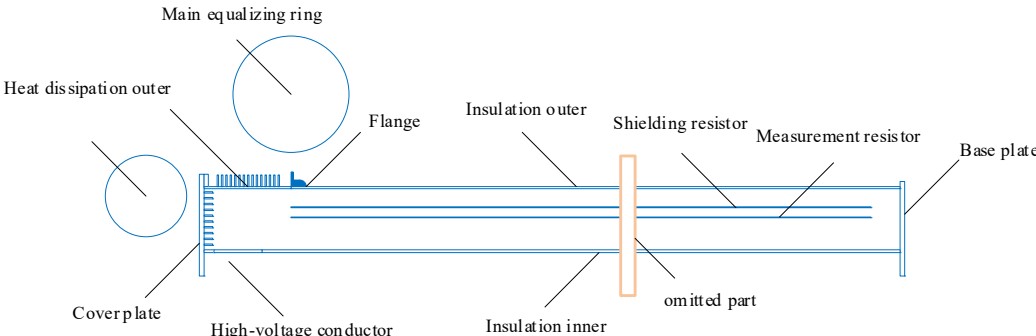

**Figure 1.** Finite-element model of 1100 kV DC voltage ratio standard device.

*Analysis of Calculation Results*

This paper conducted electric field distribution calculations on the 1100 kV DC voltage ratio standard device using the ANSYS finite-element simulation software. The simulation results are shown in Figure 2. The maximum field strength is concentrated around the two grading rings, indicating a significant impact of the grading rings on the electric field distribution of the DC voltage ratio standard device. The field strength is also relatively high at the flange, as it is a sharper point. The maximum field strength of the DC voltage ratio standard device with the initial parameters of the grading ring is 12,161.1348 V/cm. To prevent corona discharge and aging of the insulating sleeve, it is necessary to minimize the electric field intensity under the rated voltage. Therefore, the optimization design of the grading-ring structure parameters of the 1100 kV DC voltage ratio standard device is carried out with the maximum field strength around the standard device as the target.

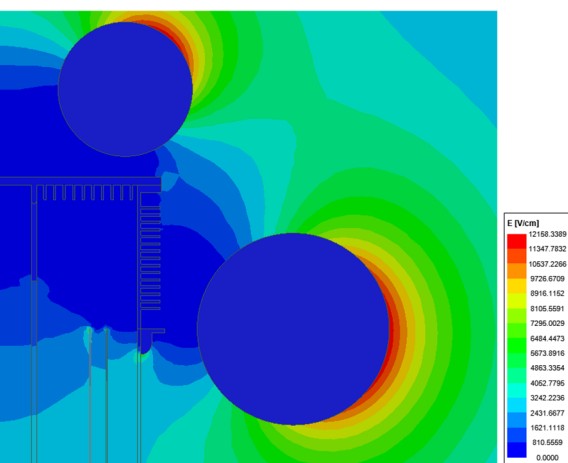

**Figure 2.** The initial DC voltage ratio standard device electric-field simulation diagram. E in the figure represents the electric field strength in various places.

### 3. The Construction of the NSD Neural Network

Secondly, based on the calculation results of the electric field, we propose a neural network named NSD to fit the relationship between the structural parameters of the grading ring and the electric field intensity. The NSD neural-network model structure consists of four fully connected layers. The input layer comprises six neurons, representing the normalized feature parameters of the main grading ring $R$, $H$, $L$ (inner radius, distance from center to ground, and distance from center to symmetry axis) and the auxiliary grading ring $R'$, $H'$, $L'$ (inner radius, distance from center to ground, and distance from center to symmetry axis). The first fully connected layer serves as the feature extraction stage, extracting input grading-ring parameter information for parameter augmentation. The second-to-fourth fully connected layers constitute the feature learning and prediction stage, employing parameter fusion techniques to refine and reuse network features, reducing the time cost of model training and inference, and enhancing feature representation capability.

The normalization formula is given by:

$$X_{norm} = \frac{X - X_{\min}}{X_{\max} - X_{\min}}, \tag{5}$$

In the equation $X$ represents the raw data, $X_{\min}$ is the minimum value in the dataset, $X_{\max}$ is the maximum value in the dataset, and $X_{norm}$ normalized is the normalized data.

For deep neural networks, when using the Sigmoid function for backpropagation, the problem of vanishing gradients often occurs, making it difficult to train deep neural networks. On the other hand, the ReLU function can alleviate the issue of overfitting. Therefore, the ReLU function is chosen as the activation function for the input and output layers.

The ReLU function is given by:

$$f(x) = \max(0, x), \tag{6}$$

The established neural network can capture the mapping relationship between the parameters $R$, $H$, $L$, $R'$, $H'$, $L'$ and $E$. The number of nodes in each layer gradually increases and then decreases, according to powers of 2. Data is first encoded and then decoded, and through further experimental comparison the selection of the optimal number of layers in the model is determined. The results of the NSD neural network are shown in Figure 3.

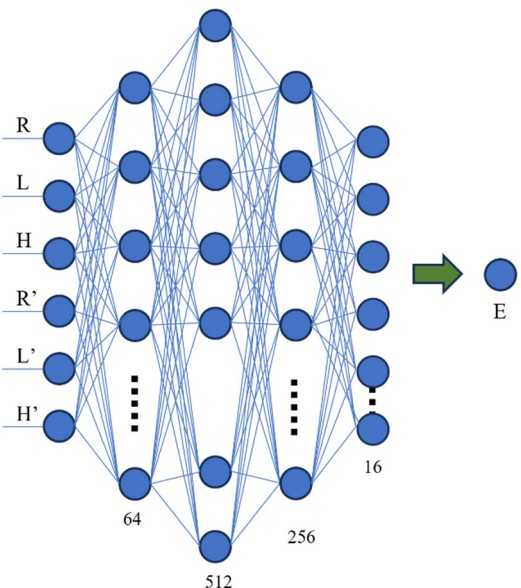

**Figure 3.** NSD Neural network architecture diagram. The dots represent neurons, the numbers below represent the number of neurons, and the lines between the dots represent full connections.

## 4. Simulation Case Validation

### 4.1. Construction of the Dataset

When optimizing the structure of the grading ring in the 1100 kV DC voltage ratio standard device, a significant field strength is concentrated on the surface of the grading ring during simulation. Therefore, the maximum field strength on the surface of the grading ring is chosen as the objective function. The structural parameters of the grading ring are considered as inputs. The 1100 kV DC voltage ratio standard device includes the main grading ring and the auxiliary grading ring, so there are six input variables, and the mapping relationship is as follows:

$$E = F(R, H, L, R', H', L'), \tag{7}$$

The optimized structure is as shown in Figure 4. The range of grading-ring structure and position parameters is shown in Table 1.

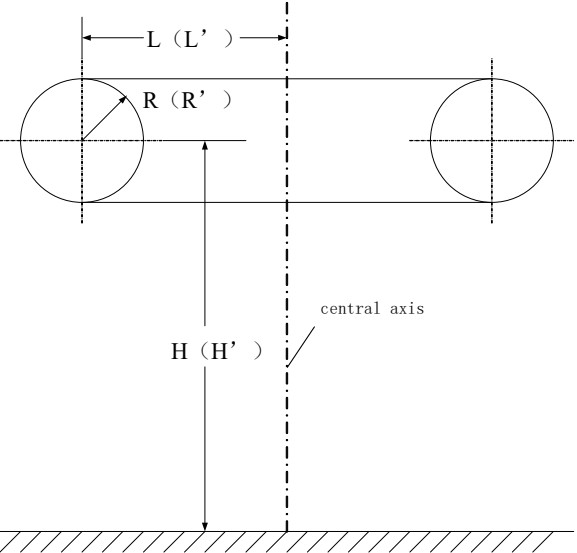

**Figure 4.** Optimized structure schematic diagram.

**Table 1.** The range of grading-ring structure and position parameters.

| The Structural Parameters | The Range of Values/mm |
|:---:|:---:|
| R | 400–500 |
| H | 7200–7350 |
| L | 1250–1400 |
| R′ | 550–800 |
| H′ | 8150–8300 |
| L′ | 550–800 |

To achieve the electric field optimization of the 1100 kV DC voltage ratio standard device, this paper first created a dataset of the maximum electric field strength for different grading-ring structures. The dataset is derived from ANSYS simulation results, where the range of grading-ring structural parameters during simulation is shown in Table 1. The dataset consists of 4096 data samples, which are split into training, validation, and testing sets, with a ratio of 8:1:1.

*4.2. Experimental Setup*

The experiment was conducted on the Ubuntu 20.04 operating system, using Python version 3.8.0, CUDA version 11.2, and training and testing were performed on the deep learning framework based on PyTorch 1.8. Two NVIDIA GeForce RTX 3090-24G GPUs were utilized for acceleration during the training process. The training was configured with 100 epochs, a batch size of 32, an initial learning rate of 0.001, and the Adam optimizer for parameter optimization.

*4.3. Experimental Results and Analysis*

This paper conducted comparative training of various models in the same experimental environment, including ridge regression [17], decision tree regression [18], random forest regression [19], gradient boosting regression [20], K-nearest neighbors regression [21], and neural network regression.

When evaluating the performance of the models, the following four metrics were selected: mean absolute error (*MAE*), mean squared error (*MSE*), root mean squared error (*RMSE*), and mean percentage error (*MPE*). The formulas for these metrics are as follows:

$$MAE = \frac{1}{n} \sum_{i=1}^{n} |y_i - \hat{y}_i|, \tag{8}$$

$$MSE = \frac{1}{n} \sum_{i=1}^{n} (y_i - \hat{y}_i)^2, \tag{9}$$

$$RMSE = \sqrt{MSE}, \tag{10}$$

$$MPE = \frac{1}{n} \sum_{i=1}^{n} (\frac{y_i - \hat{y}_i}{y_i}) \times 100, \tag{11}$$

The training results are shown in Table 2.

**Table 2.** The comparative training results for various models are presented.

| Different Approaches | MAE | MSE | RMSE | MPE |
|:---:|:---:|:---:|:---:|:---:|
| Ridge Regression | 31.4276 | 1963.4010 | 44.3100 | 0.2977% |
| Decision Tree Regression | 32.2879 | 2649.9785 | 51.4779 | 0.1505% |
| Random Forest Regression | 23.5440 | 1393.8218 | 37.3339 | −0.0014% |
| Gradient Boosting Regression | 23.4262 | 1365.6969 | 36.9553 | 0.1703% |
| K-Nearest Neighbors Regression | 28.8105 | 1628.9967 | 40.3608 | 0.4186% |
| Neural Network Regression | 22.1815 | 1119.1966 | 33.4544 | 0.1196% |

From Table 2, it can be observed that the MAE, MSE, and RMSE of the neural network regression method are the smallest among the six regression methods. The MPE of random forest regression is relatively small compared to neural network regression, but the other three indicators are larger than neural network regression. Therefore, it can be considered that neural network regression is the optimal method among the six regression methods.

The comparison between the predicted electric field values and the real values for each model is shown in Figure 5.

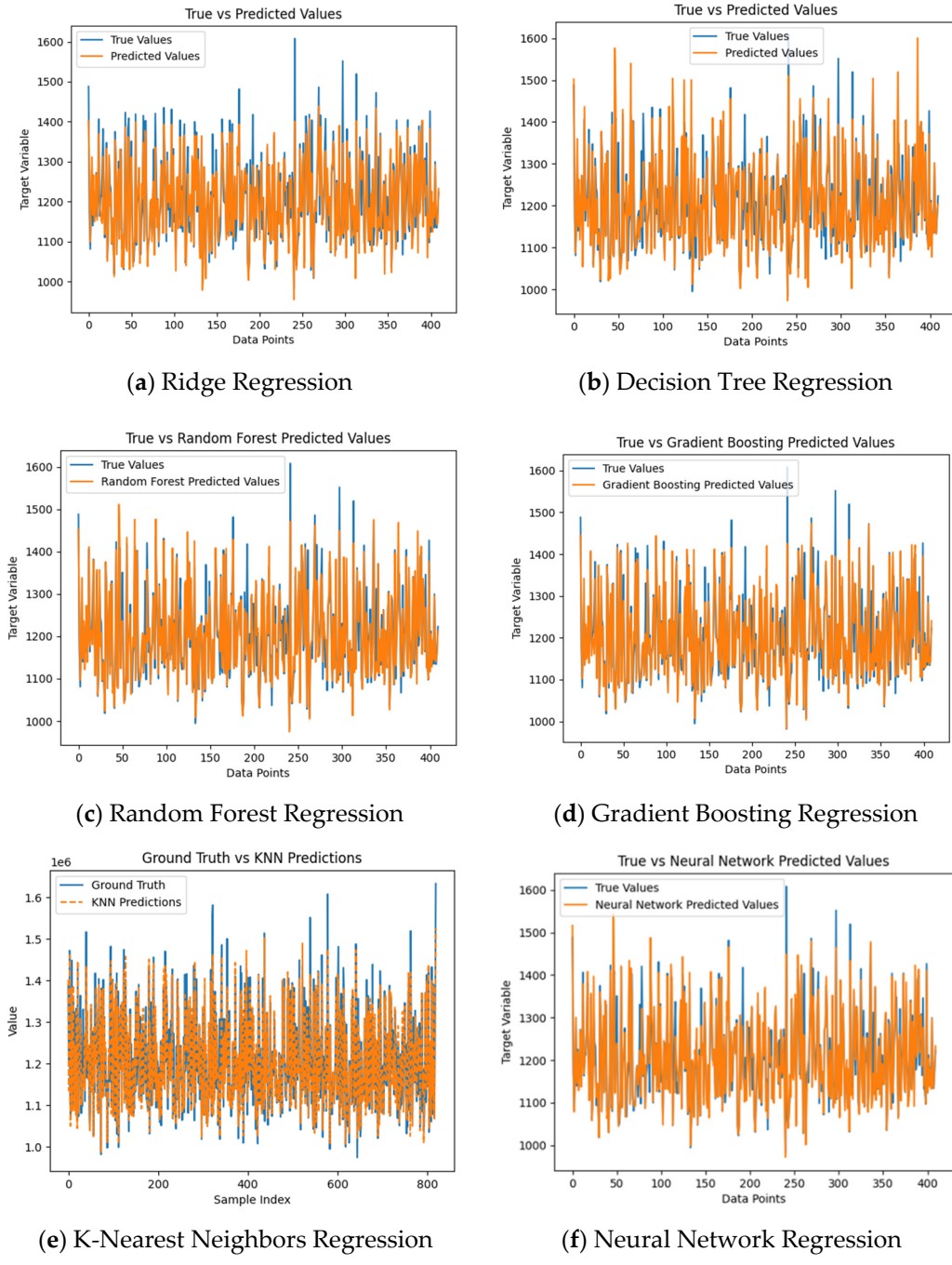

(**a**) Ridge Regression      (**b**) Decision Tree Regression

(**c**) Random Forest Regression      (**d**) Gradient Boosting Regression

(**e**) K-Nearest Neighbors Regression      (**f**) Neural Network Regression

**Figure 5.** Comparison chart of predicted values and actual values for each model. (**a**) Ridge Regression; (**b**) Decision Tree Regression; (**c**) Random Forest Regression; (**d**) Gradient Boosting Regression; (**e**) K-Nearest Neighbors Regression; (**f**) Neural Network Regression.

The above figure shows that compared to the other five traditional machine learning regression methods, neural network regression has the closest match between predicted

and actual values. It indicates that neural network regression has the best fitting effect on the electric field strength.

The number of fully connected layers was changed, and comparative training of neural network models with different numbers of fully connected layers was conducted in the same experimental environment. The experimental results are shown in Table 3.

**Table 3.** The comparative training results for various models are presented.

| Number of Fully Connected Layers | *MAE* | *MSE* | *RMSE* | *MPE* |
| --- | --- | --- | --- | --- |
| 1 layer | 43.6270 | 3124.4137 | 55.8965 | −0.0165% |
| 2 layers | 25.4261 | 1527.3979 | 39.0819 | 0.0754% |
| 3 layers | 23.3974 | 1252.3595 | 35.3887 | −0.1283% |
| 4 layers | 20.6108 | 1213.2336 | 34.8315 | 0.1232% |
| 5 layers | 24.2319 | 1237.6127 | 35.1797 | −0.3209% |
| 6 layers | 22.5858 | 1229.7693 | 35.0681 | 0.4705 |

From Table 3, it can be observed that as the number of fully connected layers increases from 1 to 4, MAE, MSE, and RMSE gradually decrease, indicating a reduction in the model's prediction errors. Although MPE slightly increases, overall the model's errors decrease. However, as the number of fully connected layers continues to increase, MAE, MSE, and RMSE instead increase, suggesting an increase in prediction errors. This may be due to the occurrence of overfitting in the model. Therefore, the neural network model with four layers of fully connected layers is chosen as the final model in this study.

The loss curve of the neural network training with four layers of fully connected layers is shown in Figure 6.

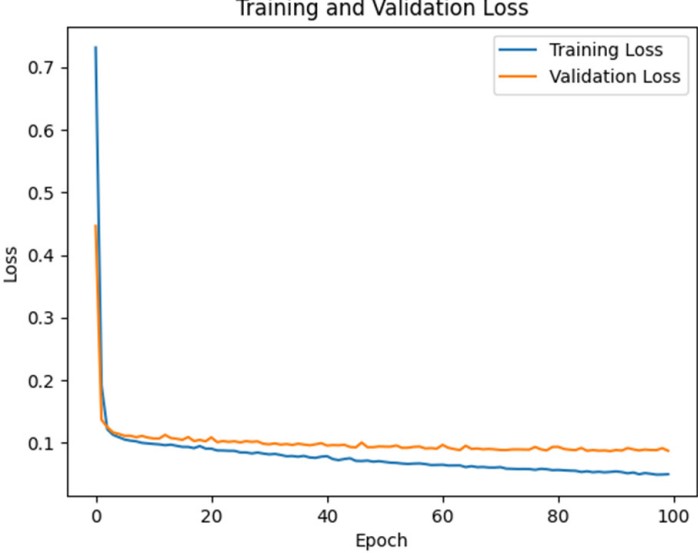

**Figure 6.** Loss Curve.

Figure 6 shows that when the number of fully connected layers is four, the model has a low loss value, and between 90 and 100 s, the model is in a stable and convergent state.

## 5. Optimization of Grading-Ring Structure Based on GWO Algorithm

### 5.1. GWO Algorithm

Finally, we optimized the structural parameters of the grading ring through the grey wolf optimization algorithm. The grey wolf optimization (GWO) algorithm is an optimization algorithm inspired by the hunting behavior of grey wolves [22,23]. In this algorithm, the fitness value is calculated for each variable combination, and the optimal

solution, second-best solution, third-best solution, and other solutions are determined through non-dominated solution sorting [24]. These solutions are defined as alpha wolf, beta wolf, delta wolf, and other wolves, respectively. The distance between a wolf and its prey can be calculated by the following formula:

$$\begin{cases} D = |C \cdot X_p(t) - X(t)| \\ C = 2\eta_1 \end{cases},$$ (12)

The formula is as follows: $D$ is the distance vector, $t$ is the iteration number, $X_p$ is the prey's position vector, $X$ is the grey wolf's position vector, $C$ is the swing coefficient, and $\eta_1$ is a random number between 0 and 1.

The formula for the position of a grey wolf from the t-th iteration to the (t + 1)-th iteration is given by:

$$X(t+1) = X_p(t) - Q \cdot D,$$ (13)

where $Q$ is the convergence coefficient.

$$Q = 2q\eta_2 - q,$$ (14)

where $q$ is a constant and $\eta_2$ is a random number between 0 and 1.

When searching for optimization, the position of the wolves at time $t$ is unknown. It needs to be determined by the positions of the α-wolf, β-wolf, and δ-wolf. Therefore, the position update for each wolf is implemented by the following formula:

$$\begin{cases} X_1 = X_\alpha(t) - A_1 \cdot D_\alpha \\ D_\alpha = |J_1 \cdot X_\alpha(t) - X(t)| \end{cases},$$ (15)

$$\begin{cases} X_2 = X_\beta(t) - A_2 \cdot X_\beta \\ D_\beta = |J_2 \cdot X_\beta(t) - X(t)| \end{cases},$$ (16)

$$\begin{cases} X_3 = X_\delta(t) - A_3 \cdot D_\delta \\ D_\delta = |J_3 \cdot X_\delta(t) - X(t)| \end{cases},$$ (17)

$$X(t+1) = (X_1 + X_2 + X_3)/3,$$ (18)

where $X_1$, $X_2$ and $X_3$ are the position vectors of the three new wolves, and $J_1$, $J_2$, and $J_3$ are constants.

When all wolf positions are updated, the optimization of the position vectors for the t-th iteration is completed [25,26].

### 5.2. Choice of the Objective Function

The goal of optimizing the grading ring is to find a set of parameters that minimize the surface electric field of the grading ring, i.e.,:

$$\min(E_{max}) = \min(F(R, H, L, R', H', L')),$$ (19)

The specific approach involves first constructing a neural network model capable of fitting the simulation of the electric field in the 1100 kV DC voltage ratio standard device. Subsequently, optimization calculations are performed to select the grading-ring structure parameters that minimize the electric field intensity.

The optimization variables selected include the inner radius of the main grading ring, the distance from the center to the ground and the distance from the center to the symmetry axis, as well as the corresponding parameters for the auxiliary grading ring. The optimization goal is to minimize the maximum electric field strength between the two grading rings. The range of values for the optimization variables is defined as follows:

$$\begin{cases} 1250 \text{ mm} \leq L \leq 1400 \text{ mm} \\ 400 \text{ mm} \leq R \leq 500 \text{ mm} \\ 7200 \text{ mm} \leq H \leq 7350 \text{ mm} \\ 550 \text{ mm} \leq L' \leq 800 \text{ mm} \\ 150 \text{ mm} \leq R' \leq 280 \text{ mm} \\ 8150 \text{ mm} \leq H' \leq 8300 \text{ mm} \end{cases}, \tag{20}$$

### 5.3. Optimization Results and Analysis

The optimization step sizes for parameters $R$, $H$, $L$, $R'$, $H'$, and $L'$ are set as [0.05, 0.01, 0.05, 0.05, 0.01, 0.05]. After 300 iterations, the GWO algorithm obtains the parameter values and optimized electric field strength for the three best solutions, as shown in Table 4.

**Table 4.** The parameter values and optimized electric field strength.

| Schemes | L /mm | R /mm | H /mm | L' /mm | R' /mm | H' /mm | E /V/mm |
|---|---|---|---|---|---|---|---|
| Scheme 1 | 1550 | 290 | 7300 | 900 | 290 | 8150 | 9988.7482 |
| Scheme 2 | 1350 | 500 | 7350 | 750 | 249 | 8250 | 10,392.8254 |
| Scheme 3 | 1350 | 500 | 7350 | 820 | 220 | 8250 | 11,023.7231 |

Comparison reveals that Scheme 1 has the minimum electric field intensity and is selected as the final optimization solution. Based on these parameters, electric field simulation was conducted, and the simulation results are shown in Figure 7.

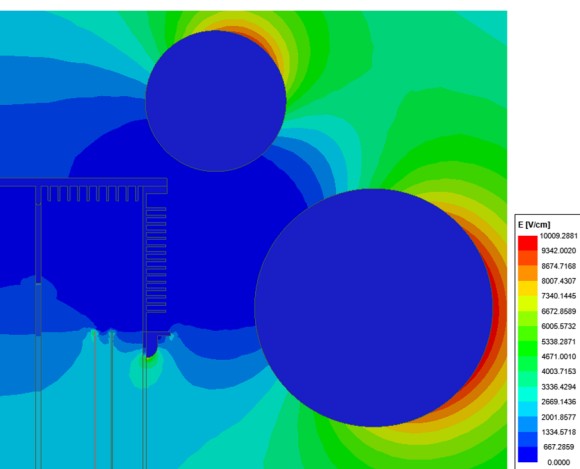

**Figure 7.** Electric field simulation diagram after optimization in Scheme 1. E in the figure represents the electric field strength in various places.

From Figure 7, it can be observed that the finite-element simulation results under the optimal grading-ring structure parameters have a small deviation compared to the predicted output results, indicating high accuracy. Moreover, the maximum electric field strength after optimization is reduced to 10,009.2881 V/cm, which is a 17.69% decrease compared to the initial maximum electric field strength.

## 6. Discussion

This paper conducted simulation calculations of the electric field of the 1100 kV DC voltage ratio standard device using ANSYS software. It designed the NSD neural network and applied the GWO algorithm to optimize the grading-ring structure of the standard device. However, there are still some shortcomings:

1. The modeling of the 1100 kV DC voltage ratio standard device in ANSYS can be further refined. The standard device is composed of thousands of resistors and

is not entirely a three-dimensional axisymmetric model. However, for the sake of simplifying calculations and reducing modeling difficulty, this paper regarded it as a three-dimensional axisymmetric model and treated serial resistors as cylinders.

2.  This paper only optimized the structural parameters of the grading ring using the grey wolf optimization algorithm, and the results were relatively ideal. However, it did not compare results with other optimization algorithms. There may be room for further improvement in the structural parameters of the grading ring.

## 7. Conclusions

This paper, based on finite-element simulation, calculated the electric field distribution around the 1100 kV DC voltage ratio standard device. The NSD neural network was designed, and the GWO algorithm was applied to optimize the grading-ring structure of the standard device. The conclusions are as follows:

The simulation model for the 1100 kV DC voltage ratio standard device was established, and the electric field was calculated. The maximum field strength under the initial grading-ring parameters was 12,161.1348 V/cm, which may lead to corona discharge and aging of the insulating sleeve.

During the design, the neural network model was compared with five other regression models. The neural network model showed the smallest MAE, MSE, and RMSE, indicating the best regression performance. When comparing neural network models with different numbers of fully connected layers, the model with four layers performed the best. Therefore, a neural network model with four fully connected layers was selected.

The GWO algorithm was employed for optimization, resulting in the optimized parameters for the main grading ring: an inner radius of 490 mm, distance from the center to the ground of 7300 mm, and distance from the center to the symmetry axis of 1550 mm. For the auxiliary grading ring the parameters are the following: an inner radius of 290 mm, distance from the center to the ground of 8150 mm, distance from the center to the symmetry axis of 900 mm. The maximum electric field strength obtained was 10,009.2881 V/cm, representing a 17.69% reduction compared to the initial grading-ring parameters.

**Author Contributions:** Conceptualization, W.Z. and Y.D.; methodology, Y.G.; software, M.H. and Y.G.; validation, W.Z., Y.G. and M.H.; formal analysis, L.Q.; investigation, W.Z. and K.L.; resources, Y.D. and Z.B.; data curation, L.Q.; writing—original draft preparation, Y.G.; writing—review and editing, Y.D. and Z.B.; visualization, Y.G.; supervision, W.Z. and K.L.; project administration, Z.B. and Y.D.; funding acquisition, W.Z. All authors have read and agreed to the published version of the manuscript.

**Funding:** This research was funded by the State Grid Corporation of China Headquarters Science and Technology Project: Key Technology Research on Self-calibration and On-site Transfer of 1100 kV DC Voltage Ratio Magnitude, grant number No. 5700-202320616A-3-2-ZN.

**Data Availability Statement:** Restrictions apply to the datasets. The datasets presented in this article are not readily available because the data are part of an ongoing study; further inquiries can be directed to the corresponding author.

**Acknowledgments:** This paper benefited from the efforts and guidance of mentor Qin Liang, mentor Liu Kaipei, He Min, and the dedication and efforts of Gao Yin, Duan Yuqing, and Bian Zhigang of State Grid Anhui Electric Power Company Limited.

**Conflicts of Interest:** Authors Wanjun Zhu and Min He are students at Wuhan University, Authors Liang Qin and Kaipei Liu are employed by the Wuhan University, Authors Yin Gao, Yuqing Duan, Zhigang Bian are employed by the company State Grid Anhui Electric Power Company Limited. The paper reflects the views of the scientists and not the company. The remaining authors declare that the research was conducted in the absence of any commercial or financial relationships that could be construed as a potential conflict of interest.

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
