# Peer review of "Optimal Design of Voltage Equalization Ring for the 1100 kV DC Voltage Proportional Standard Device Based on the Nation Standard Device Neural Network and Grey Wolf Optimization Algorithm"

_electronics, doi:10.3390/electronics13071308_

Round 1

Reviewer 1 Report

Comments and Suggestions for Authors

This Proposed manuscript titled, "Optimal design of voltage equalization ring for 1100kV DC 2 voltage proportional standard device based on NSD neural network and Gray Wolf Optimization algorithm”, the idea and proposed work is  interesting. I would like to suggest few comments :

1. In the Abstract authors should mention about the software used to implement this work.

2. Authors must include a section to section to show the core contribution of this paper.

3. A discussion on the possible limitations of their work may be included to enhance the quality and ethical fairness of their work.

4. The quality of Fig 1 and Fig. 6 must be improved.

5.Authors must include a section for all the abbrevaitions

·         

·      

Comments on the Quality of English Language

Moderate editing of English language required

Reviewer 2 Report

Comments and Suggestions for Authors

1.    Maintain a gap spacing between the unit and magnitude throughout the manuscript (Eg. 1100 kV, 12161.1348 V/cm).

2.    Maintain the same spelling for Grey Wolf Optimization throughout the manuscript (In the title, it is mentioned as Gray and other places as Grey).

3.    ANSYS should be mentioned in capitalized and not Ansys.

4.    The authors have mentioned about the importance of sigmoid and ReLU as an activation function? Why not hyperbolic tangent activation function? Explain.

5.    On what basis did the authors select four connected layers in the NSD neural network model?

6.    How did the authors choose the ratio of training, validation and testing sets to be 8:1:1 and if this ratio is changed will it affect the accuracy of the results?

7.    What was the relative error set in GWO algorithm to obtain the optimization results?

Reviewer 3 Report

Comments and Suggestions for Authors

Couple of pointers / explanations that I'd like to see addressed in the final revision - 

1. Neural networks and heuristic algorithms such as the GWO are used in highly non-linear and non-convex problem statements. The model used to train the NSD used a simplfied verison of the ring design to calculate the field strength. How much an improvement are we seeing using these methods ? Is the electric field calculation really that non-linear and complex that warrants an NSD and GWO to optimize ? Could you add to that ? Obviously the NSD is highly non linear, so using either genetic algorithm or GWO or any other heuristic algorithm is fair. 

2.  I guess this follows the previous question in a way. To train the NSD, over 4000 data points were simulated and while I commend that the NSD training showed good results, how does this compare to say going over those 4000+ datapoints and identifying minima in thst data set ? I guess the question harks back to how non-convex was that data set ? Were there multiple local minima's ? 

3. The authors cite that exhaustive search is inefficient and difficult to find optimum parameters. Isn't the very definiton of exhaustive search to systemtically "exhaust" every possible option and identifiy the optimum point ? While it certainly is time conusuming and thus inefficient , how does it not make finding optimum parameters difficult ?

4. Was the NSD tested and verified on data outside the bounds of the test data according Table 1 ? Was the NSD designed as a tool to identify the electric field in this particular shape or does it only work for this shape and particular parameters ?

5. I guess one of the skepticisms I have is that the NSD seems to be for a specific shape and if the designer wanted to say add another design parameter or so, would have to train the model again with a new set FEA simulations for data points. Given all that effort, how much faster or better is this method compared to just going for a full exhaustive search ?  

Comments on the Quality of English Language

The quality of the language is fairly good, I dont see any major issues, but I would add that at times the wording seems extraneous and neccasrily long.

Round 2

Reviewer 1 Report

Comments and Suggestions for Authors

Iwish Good luck to authors

Comments on the Quality of English Language

Minor editing of English language required

Reviewer 2 Report

Comments and Suggestions for Authors

The paper is well written after the reviewer suggestions and is accepted

Reviewer 3 Report

Comments and Suggestions for Authors

The issues that I had brought up have all been addressed, so no concerns or additional pointers